# Prevalence of the Mutations Responsible for Glanzmann Thrombasthenia in Horses in Brazil

**DOI:** 10.3390/ani9110960

**Published:** 2019-11-13

**Authors:** Raíssa O. Leite, Júlia F. Ferreira, César E. T. Araújo, Diego J. Z. Delfiol, Regina K. Takahira, Alexandre S. Borges, Jose P. Oliveira-Filho

**Affiliations:** 1São Paulo State University (Unesp), School of Veterinary Medicine and Animal Science, Department of Veterinary Clinical Science,18618-681 Botucatu, Brazil; raissaleitevet@hotmail.com (R.O.L.); julia.franco.ferreira@gmail.com (J.F.F.); cesararaujovet@hotmail.com (C.E.T.A.); regina.takahira@unesp.br (R.K.T.); alexandre.s.borges@unesp.br (A.S.B.); 2School of Veterinary Medicine, Universidade Federal de Uberlândia, 38405-320 Uberlândia, Brazil; djzdelfiol@ufu.br

**Keywords:** genetic disease, prevalence study, fibrinogen receptor, epidemiology

## Abstract

**Simple Summary:**

Hereditary bleeding disorders occur in different species due to mutations in genes coding specific hemostatic proteins leading to alterations in their synthesis, or to the production of non-functional proteins which leads to impairment of hemostasis. Some of these disorders have been described in horses, i.e., Von Willebrand disease (VWD), hemophilia A, and Glanzmann’s thrombasthenia (GT). GT is an inherited disease characterized by hemorrhage and has been described in different species including horses of varied breeds (Thoroughbred, Standardbred, Oldenburg, Peruvian Paso, and Quarter Horse). There are two different mutations described in horses a single guanine to cytosine substitution (CGG for CCG) and a 10 base pair deletion in the *ITGA2B* gene.

**Abstract:**

Glanzmann’s thrombasthenia (GT) is an autosomal recessive inherited disorder characterized by changes in platelet aggregation, leading to hemorrhage and epistaxis. To date, two independent mutations have been described in horses and associated with this disorder, a point mutation (c.122G > C) and a 10-base-pair deletion (g.1456_1466del) in the Integrin subunit alpha2β gene (*ITGA2B*) of horses of different breeds (Quarter Horse, Thoroughbred, Oldenburg, and Peruvian Paso). *ITGA2B* codifies the αIIb subunit of the αIIbβ3 integrin, also termed platelet fibrinogen receptor. Horses with GT have been diagnosed in the USA, Canada, Japan, and Australia. However, there are no studies on the prevalence of GT in horses. The aim of this study is to evaluate the prevalence of the mutations responsible for GT in horses in Brazil. A total of 1053 DNA samples of clinically healthy Quarter Horse (n = 679) and Warmblood horses (n = 374) were used. DNA fragments were amplified by PCR and sequenced. The genotype of each animal was analyzed and compared to the nucleotide sequence of the *ITGA2B* gene found on GenBank^TM^. There were no carriers in the analyzed samples, that is, all animals tested were wild type. Therefore, under the conditions in which this study was carried out, it can be inferred that GT seems to be extremely rare in the population of Quarter Horses and Warmbloods in Brazil, although it is not possible to affirm that there are no horses carrying mutated alleles in Brazil.

## 1. Introduction

Hereditary bleeding disorders occur due to mutations in genes coding specific hemostatic proteins, leading to alterations in their synthesis, or to the production of non-functional proteins [1]. Some of these disorders have been described in horses, i.e., Von Willebrand deficiency (VWD) [2], hemophilia A [3], and Glanzmann’s thrombasthenia (GT) [4,5,6,7,8,9].

GT occurs due to quantitative or qualitative changes in the glycoprotein IIb-IIIa complex, also known as αIIbβ3 integrin or fibrinogen receptor, which is found on the surface of the platelets and has an essential role in platelet aggregation and clot retraction [1,5,6,10]. Mutations involving the genes that encode the subunits αIIb (*ITGA2B*) or β3 (*ITGB3*) can result in GT. In animal domestic species, mutations have only been noted in the *ITGA2* gene [5,6,9], whereas in humans’, mutations have been documented in both genes [10]. Clinically GT is characterized by cutaneous, mucosal, and gastrointestinal hemorrhages, and epistaxis is the most frequent clinical sign [6]. Laboratory tests of affected animals demonstrate normal coagulation screening tests, normal platelet counts, and normal Von Willebrand factor antigen levels, with clot retraction and platelet aggregation responses impaired; and also, a reduction in αIIbβ3 integrin indicated by flow cytometry [5,6,9,11]. This disorder has been described in Thoroughbred [4,7,8,11,12], Peruvian Paso [10] Standardbred [12], Oldenburg [13], and Quarter Horse breeds [8].

In horses, GT has an autosomal recessive inheritance pattern [5,6,9,11,13] and has been associated with two independent mutations, a point mutation (c.122G > C) [5,9] and a 10-base-pair deletion (g.1456_1466del) [6,11], in the Integrin subunit alpha2β gene (*ITGA2B*) of horses of different breeds (Quarter Horse [5,6], Thoroughbred [5], Oldenburg [9] and Peruvian Paso [11]). The point mutation leads to the substitution of a proline for an arginine in a highly conserved region of the encoded protein [5], and the deletion mutation leads to a lack of splicing, and inclusion of a premature stop codon 50 bp downstream of the mutation in the incompletely spliced messenger ribonucleic acid (mRNA) [6]. In horses, studies of the prevalence of theses mutated alleles have not been performed, therefore, the aim of this study was to evaluate the prevalence of the mutations describe as responsible for Glanzmann’s Thrombasthenia in horses in Brazil.

## 2. Materials and Methods

This study was approved on 21 January 2019 by the Institutional Animal Care and Use Committee (262/2011-CEUA-UNESP) and samples were collected under a strict confidentiality agreement to ensure the anonymity of establishments, owners and animals. Since the allele frequencies of the two mutations are unknown, we used the anticipated frequency of heterozygotes of 50% [14], a population of Quarter Horse (QM) (500,000) and Warmblood (WB) horses (14,000) registered in Brazil, using a 5% margin of error, and 95% confidence interval to calculate the sample size (OpenEpi software). The sample size recommended for QM was 384, and 374 for the WB horses.

A total of 1053 horses DNA samples were used in this study. These samples were obtained from a genetic material database belonging to the Laboratory of Molecular Biology of the Veterinary Clinic at São Paulo State University (Unesp), School of Veterinary Medicine and Animal Science Botucatu/Brazil. This genetic database is composed by DNA samples (stored at −80 °C) which were extracted from whole blood samples obtained from adult horses (males and females), duly registered in the breeds associations (Brazilian Association of Quarter Horses Breeders and Brazilian Sport Horse Association). The samples from QHs were collected from 41 different farms and the WB samples from seven different farms or horses training centers.

Genotype analysis was performed using specific primers previously described for detection of the *ITGA2B* g.1456_1466del mutation [6] and for detection of the *ITGA2B* c.122G > C mutation [9]. Polymerase chain reactions were performed in a total volume of 25 μL, which contained 20 ng/μL (2.5 μL) of template DNA, 0.3 μM each forward and reverse primer, 12.5 μL of GoTaq^®^ Green PCR Master Mix (Promega©, Madison, WI, USA), and 8.5 μL of nuclease-free water. In addition, a no-template control reaction was performed to check for the possible presence of contamination in the PCR preparations. The amplification conditions were as follows: initial denaturation at 95 °C for 2 min, followed by 40 cycles of denaturation at 95 °C for 30 s, 64 °C for 30 s, and 72 °C for 1 min, and final extension at 72 °C for 5 min. Amplicons (241 bp for the deletion mutation and 359 bp for the substitution mutation) were analyzed via 1.5% agarose gel electrophoresis, purified, and subjected to Sanger direct sequencing. The obtained sequences and the electropherograms were analyzed using Geneious^®^ 10.0.9 (Biomatters©, Auckland, New Zealand).

## 3. Results and Discussion

A total of 1053 horses DNA samples were analyzed, i.e., 679 QH (483 females and 196 males) and 374 WB horses (203 females and 171 males). All horses assessed in this study were identified as wild-type for both mutations. That is, there were no carriers in these populations.

The present study is the first report of the prevalence of the mutations (c.122G > C and g.1456_1466del) responsible for GT in horses. Although this disease has not been described in Brazil, studies in other countries may contribute to define the importance of GT for this species, since hemorrhage is a common clinical sign in horses and its genetic causes remain poorly elucidated [15]. In addition, the knowledge of the allele frequency of a genetic disease in a population and the identification of homozygous or heterozygous animals, enables the establishment of the impact and importance of these disorders to the species, especially for specific breeds [16].

Other studies on prevalence of different diseases with hereditary patterns that affect horses, but no related to hemostasis alterations, have been performed in Brazil: Hereditary equine regional dermal asthenia [17], Hyperkalemic periodic paralysis [18], Type 1 polysaccharide storage myopathy mutation [18], Malignant hyperthermia [19], Glycogen branching enzyme deficiency [20], and Warmblood fragile foal syndrome [21]. Many of these studies demonstrated similar results as found in other countries, such as the USA [16,22]. This may be related to the fact that lineages of the main horses of Brazil are often closely correlated with American herds, and it is suggested that the same situation for GT may be found in other countries, since the USA is the largest exporter of breeds like QHs, and Brazil is the fifth largest breeder.

The prevalence of GT in humans is higher in populations where inbreeding is commonly practiced and may have a frequency similar to other hemostatic diseases, such as hemophilia and VWD [10]. Considering that inbreeding is a common practice in horses [23], there may be differences between the frequencies of occurrence of GT in some horses’ populations.

## 4. Conclusions

In summary, the mutations responsible for GT were not detected in this prevalent in the Brazilian horse population. Under the conditions in which this study was carried out, it can be inferred that although it is not possible to affirm that GT do not occurs in horses in Brazil, this disease seems to be extremely rare in the population of QH and WB.

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
