# Peer review of "Prevalence of the Mutations Responsible for Glanzmann Thrombasthenia in Horses in Brazil"

_animals, 2019, doi:10.3390/ani9110960_

Round 1

Reviewer 1 Report

The authors have determined the "Prevalence of the mutations responsible for Glanzmann Thrombasthenia in horses in Brazil". 

Although the study methodology is interesting, there are some points that have not been clear to this reviewer.

-What inclusion criteria have been taken into account when selecting animals for the study?. Samples of QH were collected from 41 farms and  WB from 7 different farms (Lines 82-83). This reviewer does not understand the meaning of this phrase "All horses assessed in this study were identified as homozygous for the wild-type allele for both mutations" (lines: 98-99). Do the authors know the inbeeding of these animals?. 

-As expressed in the lines: 101-102 there are no documented cases of GT in equidae in Brazil. So I do not understand based on what the authors determined the prevalence of this pathology?. This reviewer thinks that the study would have been more interesting in those equine populations in which some family relatives have previously developed the pathology, in which case it is necessary to know the prevalence in these populations. 

In the same way, if there are no documented cases of GT in Brazil, the keywords "Epistaxis" and "Hemorrhage" should be deleted from the text.

-This reviewer thinks that any other pathology in the horse should be removed from the text (for example line: 107-111) since that although they are of genetic origin, are pathologies not related to hemostasis.

Reviewer 2 Report

This is a nice, clear manuscript with sound scientific merit. There are only a few changes -minor ones. 

Line 22 change to "Although hemorrhage is the clinical sign frequently found in the equine."

Line 27 change "has" to "have"

Line 31 add the word "the" in front of "USA"

Line 36 change to "There were no carriers"

Line 37 change the first "animals" to "samples"

Line 38 please put the "although it is not possible to affirm..." sentence after the "GT seems to be extremely rare..."

Line 53 add an s to test (should be tests)

Line 56 change cytometric to cytometry

Line 58 change to "GT has an autosomal recessive inheritence pattern..."

Line 65 put mutated before alleles.

Line 66 remove the phrase "in the world"

Line 69 change approved in to approved on

Line 84 change describe to described

Line 100 add "the" in front of prevalence

Reviewer 3 Report

Lines 48-57: You should consider providing a bit of clarity in this paragraph as to what species you are referring to. For example, the general statements made about the molecular basis for GT refer to what we know in people (mutations have been documented in both genes. In veterinary species, mutations have only been noted in ITGA2B gene. You might want to clarify that a bit.

Lines 104-106: Some grammar changes are needed in this sentence, even with the most recent changes.

Round 2

Reviewer 1 Report

The manuscript reading has improved.

Line 36: Remove ":"

Lines 38 and 52: This reviewer still does not understand why a prevalence study is done in clinically healthy horses.

Line 58:  The autosomal recessive term is duplicated. Please remove.

This manuscript is a resubmission of an earlier submission. The following is a list of the peer review reports and author responses from that submission.

Round 1

Reviewer 1 Report

Most changes have been made successfully. I have a few additional comments below.

Lines 36-37: I know this sentence was already updated, but the wording still could be more concise. Consider changing.

Line 48: “The GT” can just be “GT”

Lines 104-105: Grammar correction still needed

Reviewer 2 Report

This reviewer agrees that the mutations that the authors referred are accompanied by the change of the sequence of amino acids and may relate to the occurrence of GT. However, this reviewer does not consider that the literatures that the authors offered are enough to believe that Glannzman’s thrombasthenia (GT) is “an autosomal recessive inherited disorder” and caused only by the mutations that the authors are mentioned in the manuscript.

In many cases, genetic traits including genetic disorders are regulated by multiple genes and affected by environmental factors. Are there any data perfectly prove that the mentioned mutations are the only one factor to regulate GT occurrence? (The attached literatures by the authors do not include the data that proves the mentioned mutations are the only factor to cause GT.) If the authors want to mention GT as recessive inherited disorder by all means, at least the literatures proving that those mutations are the only one responsible DNA regions of GT occurrence by some genome-wide analyses should be attached.

To clearly state the degree of the effect of the concerned mutations in the introduction section is very important for this manuscript. If the mentioned mutations are the only cause of GT, the data included in this manuscript is valuable. However, if those mutations are not the major regulatory factor of GT, the data in the manuscript is totally worthless.

The expressions “GT is an autosomal recessive inherited disorder” and "This disorder was associated with two independent mutations" found in line 25-27 in the abstract section tend to make mislead the readers to believe that the mentioned mutations are already proven as only one factor to cause GT. Therefore the introduction section of this manuscript should be improved completely.

Reviewer 3 Report

All comments have been addressed with the exception of the comments regarding power. I understand what the authors have discussed in those points, and am happy that the conclusions are appropriate.